# Application of Inertial Sensors to Identify Performance-Relevant Parameters in Olympic Hammer Throw

**DOI:** 10.3390/ijerph19095402

**Published:** 2022-04-28

**Authors:** Stefan Tiedemann, Thorben Menrad, Kerstin Witte

**Affiliations:** Department Sports Science, Faculty of Humanities, Otto von Guericke University, 39106 Magdeburg, Germany; thorben.menrad@ovgu.de (T.M.); kerstin.witte@ovgu.de (K.W.)

**Keywords:** hammer throw, inertial sensors, motion capturing

## Abstract

The aim of this study is to find essential performance-determining biomechanical parameters in hammer throw. There is no consensus in science on this, for many reasons. Among other things, there is the high complexity of the movement in conjunction with the differences in the physical conditions of the individual athletes. The purpose was to make complete body analyses of six experienced throwers (2 × male, 4 × female). Methods: The kinematics were collected with the full body sensor suit (Moven suit from Xsens with 17 inertial measurement units (IMU)). We considered 37 literature-based parameters. By means of correlation analyses, with regard to throwing distance, and a principal component analysis (PCA), performance-relevant parameters could be found. Results: The most promising results occurred in stance times and left hand speed. These findings suggest, in accordance with other studies, that these parameters have a relevant influence on the throwing distance. Comparing acceleration and angular velocity with the throwing distance also look relevant for performance. Conclusions: Further research with a focus on technique and bigger differences in the throwing distance are necessary to obtain clearer performance relevant parameters.

## 1. Introduction

The hammer throw has been part of the Olympic Games since 1900 and since then there have been various efforts to optimize the technique of the throwers. In order to increase the performance in throwing, it is necessary to address the conditional abilities and the technique of execution. It could already be proven that improvements in strength, agility and endurance, and thus in initial velocity throwing speed, lead to an increase in the throwing distance [1,2]. Furthermore, it can be assumed that the optimization of coordinative parameters also positively influences performance. However, comparatively few empirical findings are available in this area. Up to now, mainly optical systems have been used to determine biomechanical parameters [3]. The results obtained from optical methods, which are often very time-consuming to determine, are put in relation to the throwing distance. However, since the hammer throw movement is extremely complex and individual [4], there is no consensus yet on the performance-relevant parameters.

The hammer throw is characterized by three phases: 1. Winds (arm circle swing phase), 2. Turns (full body turns), 3. Release (drop phase) [5].

The throw begins with the preparation phase (winds), the arm circle swing phase. In this phase, the hammer head is moved around the body axis by arm circle swings without the athlete rotating himself. During this movement, most of the velocity is already generated [5].

In the second phase, also referred to as the main phase (turns), the focus is on the different stance phases, distinguishing between the single-support phase (SS phase or single-leg stance phase) and the double-support phase (DS phase or double stance phase). In turns, the athlete rotates together with the hammer three or four times around the rotation axis. These rotations are crucial for the maximum terminal velocity, which together with the release angle and the release height determines the throwing distance [6]. The last phase is the release and irrelevant for the performance optimization.

Competition analyses have shown that throws with shorter DS phases lead to a longer throwing distance [7,8,9,10]. Therefore, the DS phase, in which both feet are in contact with the ground, is important for the throwing distance because the athlete uses this phase to accelerate the hammer. In contrast, the SS phase is mainly used by the athlete to pass the sports equipment and quickly put both feet back on the ground. The SS phase begins with the release of the rotating leg. In training practice, this factor should therefore be taken into account and checked as often as possible. However, Brice, Ness, Everingham, Rosemond and Judge [11] show that too much focus on shortening the SS phase could be detrimental, as this is often at the expense of balance, resulting in a reduction in hammer head terminal velocity.

In order to divide the turns into the SS and DS phases, it is important to identify the events in which the circulating leg leaves the ground (lift) and touches down again (set). Stache [12] built a first measurement method for the event identification, which is based on a contact sole. This can be used in training to quantify motion sequences with objective values.

In recent years, there have also been efforts to use existing knowledge to create real-time feedback systems to enrich technique training. For example, Murofushi et al. [4] designed a self-made system that is also capable of measuring the rotational radius of the hammer and velocity compared to a conventional video system.

In 2008, Brice et al. [13] presented a method for cable force measurement based on strain gauges. It was shown that this method is more suitable than a conventional video analysis. In further work in 2014, Brice et al. [14] calculated the hammer head velocity using this strain gauge data. The two models that were set up for this purpose, show very good results in conformance with the used camera system.

Wang et al. [15] built a wireless system based on an infrared entrainment sensor. This determines the distance of the hip to the ground, which is considered a relevant parameter by the authors. In addition, a load cell is located in the hammer cable. The data obtained in this way are sent to a terminal device, where it is directly analyzed. In a further study, Wang et al. [16] converted the system to inertial sensors and verified the obtained data with a Vicon system.

One possible way to analyze the whole hammer throwing movement is to use the Moven suit (MVN) by Xsens Technologies B.V (Netherland) [17] with the MVN Analyze software. This full-body sensor suit has not yet been used in the hammer throw, but in many other areas such as swimming, snowboard, ski jumping, running, football, etc. [18]. It offers kinetics and kinematics-like positions of 23 segments. This extremely large amount of movement data (measures directly the acceleration (m/s^2^) and angular velocity °/s), can be analyzed objectively and automatically by computer software [19].

The aim of this work is to find out which biomechanical parameters that can be measured with full-body analysis are relevant performance parameters for the hammer throw (Figure 1).

## 2. Materials and Methods

The study was conducted with six hammer throw athletes (mean age: 21.83 ± 5.35 years, mean body mass: 74.33 ± 18.71 kg, mean body height: 1.75 ± 0.11 m, two male and four female). The subjects were members of the national or youth national team of the German Athletics Association. Five athletes used a throwing technique consisting of four turns. One athlete only used three turns. Table 1 contains the characteristics of the athletes and the number of trials evaluated.

Figure 2 shows the experimental set up with two cameras in relation to the athlete. These were used for the visual feedback to better understand the movement data from the inertial sensor suit: a GoPro Hero 6 (GoPro Inc., San Mateo, CA, USA, 240 Hz) and a Sony Alpha 6300 (Sony Corporation, Tokio, Japan, 120 Hz).

The MVN motion capture system (Figure 3) consisting of a Link System and the MVN Analyze software application by the company Xsens Technologies B.V. (Netherlands) was used. The Moven suit consists of 17 MTx inertial measurement units (IMU (thresholds: 16 g and 2000°/s)). Each of them consist of three axis of accelerometer, gyroscope and magnetometer (nine degree of freedom). The data were recorded with a frequency of 240 Hz. The MVN Analyze software calculates the whole body model from this. Additionally, the raw data of the individual sensors can also be viewed. No additional filter was used. In the study, throws were measured as part of the training routine of the participating athletes. All athletes used a hammer with individual hammer weights (Table 1) and made the regular number of throws in a training session.

On measurement day, the anthropometric data were measured from each athlete to improve the quality of the biomechanical model created by the Xsens software. The athlete had to wear a lycra suit that was included in the Moven suit. IMUs, a battery and transmitter were fixed to the suit and were wired together. The IMUs belonging to both feet were fixated under the Velcro fastener of the shoes (Figure 4). The sensors of the hands were mounted to the back of the hand using tape. A glove was worn over the IMU. After preparing the athletes, the MVN System had to be calibrated. An N-Pose and Walk calibration were used. The athletes completed their own regular warm-up routine. Before each throw, the athlete was given a starting signal by the investigator. At the same time, the recording was started within the MVN Analyze software. After receiving the starting signal, the athlete performed the usual throwing routine and the suit has continuously sent all the data via WIFI to the connected computer (live feed). After the hammer was released, the recording of the MVN System was stopped. The throwing distance was manually measured by the trainer.

Measurement data were exported using a right-handed Cartesian coordinate system from MVN Analyze as an .mvnx file and processed with Matlab R2020b.

For the determination of some temporal parameters, such as the time of the last DS phase, a definition of the release point is necessary (Table 2). For this purpose, the point of maximum angular acceleration of the right hand was used. This parameter was measured with the sensor that includes a gyroscope positioned there. Figure 5 shows a typical time course angular acceleration of the right hand for an exemplary throw. The maximum peak of the angular acceleration characterizes the release point (Figure 5) and can be detected automatically from this sensor.

Statistical analyses were conducted using IBM SPSS Statistics 23.

Table 2 contains the most relevant parameters and definitions for the determination of double- and single-support phases. All the measured 37 parameters that were used for determination are displayed in Table A1. Pearson correlation coefficients for these variables were calculated for each athlete (*n* = 6) in relation to the throwing distance. Level of significance was set to 5%. Pearson correlation coefficients between 0.00–0.09 were interpreted as negligible correlations; 0.10–0.40 as weak correlations; 0.40–0.69 as moderate correlations; 0.70–0.89 as strong correlations; and between 0.90–1.00 as very strong correlations [20].

Furthermore, principal component analysis was conducted to find parameters, which load significantly to the principal components and to interpret them as influential parameters for throwing distance and throwing technique. The Kaiser-Meyer-Olkin (KMO) test for sampling adequacy and Bartlett’s test of sphericity were calculated to verify the viability of the data used to perform PCA. Scree plots justified the use of three principal components. The component matrix was rotated using the Varimax method. Parameter loads with an absolute magnitude smaller than 0.64 were ignored to ensure that parameters would only load to one principal component. Duration of turns, as well as durations of double- and single-support phases, were excluded as they are calculated as the sum of other parameters used in the PCA. Throws of all athletes using four turns were included resulting in a data set of *n* = 57 throws. Not every throw could be evaluated because some throws were invalid or landed in the net.

## 3. Results

Based on the literature research, a list of 37 possible relevant biomechanical parameters were assembled with the goal of identifying the most important parameters that probably influence throwing performance. Various algorithms are created for generating these parameters automatically and show that all parameters can be analyzed with the inertial sensor suit. The summary of all parameters together with their definitions or descriptions on how to calculate them from MVN data can be found in Table A1. Angle definitions used by MVN.

For each parameter and each athlete, the Pearson correlation coefficient related to throwing distance is calculated (Table 3). The totality of all parameters with correlations can be seen in Table A1.

The duration of turns shows four significant correlation coefficients for four athletes, three being strong and one being a moderate negative correlation. The four athletes also exhibit significant negative correlation coefficients (three moderate/one strong) for the duration of the double-support phases. Additionally, non-significant correlation coefficients for these parameters still show negative values of absolute magnitude ≥ 0.30. Furthermore, moderate correlation coefficients are found for three athletes for left hand velocity at release. Durations of individual single- and double-support phases, angular velocity at release of both hands and velocity at release of the right hand all show only two or less significant correlation coefficients. Correlation coefficients for all examined parameters are displayed in the Table A2.

In addition, all parameters were subjected to a PCA with the aim of being able to infer the relevant parameters for performance from the parameters loading on the principal components. The analysis was performed with five of the six athletes, since one athlete rotated only three times. The Kaiser-Meyer-Olkin test was 0.743 and excided the required value of 0.6. Bartlett test was highly significant (*p* < 0.001). Only principal components with values greater than 1 were considered [21,22]. Scree plots justified the extraction of three principal components that described 66.48% of variance of the system. Table 4 shows all parameters that loaded significantly on one principal component. All principal components consist of temporal parameters, joint angles and velocities of the different turns. Principal component 1 further includes spatial parameters in the form of differences in pelvis height. Angular velocities at release of both hands do not load significantly to one of the principal components and only the upper body distortion (xfactor) at the set of the circumferential foot does. Loads of the parameters on the principal components are displayed in Table 4.

## 4. Discussion and Conclusions

The present pilot study attempts to eliminate a deficit that lies therein that data collection by means of the conventional performance diagnostics in the hammer throw (e.g., video analysis) is very time consuming and does not justify the effort.

It could be shown that by means of a full-body sensor suit, performance-relevant biomechanical parameters of the hammer throw can be measured. In particular, with the Moven suit it is possible to automatically and objectively determine a large number of parameters.

This measurement system is capable of capturing additional parameters and allows the evaluation of new parameters, such as the angular velocities of each body segment. These have probably not been addressed in previous literature because the measurement methods used to date are not accurate enough.

For the determination of the stance times, the time point of release is relevant. The corresponding sensor on the hand is crucial for this. Figure 4 shows a clearly automatically detectable event for the computer software. With optical methods, the categorization of this event is subjective and can therefore influence the time point and duration of the last double-support phase. Competition analyses are usually performed with video recordings of 50 Hz [9]. A small mistake of one frame in the determination of the release already leads to a deviation of 0.02 s. A similar detecting method was used for the contact times of the rotating leg [23]. The Xsens system is working with 240 Hz, which corresponds to an accuracy of 0.004 s. The advantage of this system lies in the wired networking of all 17 sensors, which can synchronously record data. This is also the base of the whole-body model.

The data analysis revealed that various parameters have an influence on the throwing distance. Individual significant correlations occurred for each athlete. It must be taken into account that only six athletes were available for this study. These athletes additionally differed in their performance capacity (Table 1). We consider that stance times are important because they show significant correlations in four of six athletes. This shows that there is probably no parameter that has an influence on all athletes. The speed of the left hand is relevant for three of six athletes. It can be assumed a correlation with the throwing speed, which significantly influences the distance [24]. We assume that these two parameters are also crucial for other athletes.

Therefore, general conclusions can only be drawn very cautiously. As described in the introduction, previous studies have used different methods to collect biomechanical parameters. In most cases, the results obtained there cannot be considered across athletes because the number of subjects and experiments is not sufficient. In many papers, only one or two subjects with different performance levels were studied [3]. The personal best throwing distance of our most powerful athlete is just over 70 m, which is far away from the distances achieved by the world’s top athletes (actual world record: 86.74 m). In comparison with competition analyses, several throws are considered here individually and across athletes. Competition analyses often look at the best attempt of each athlete and then make comparisons between the individual athletes. For this reason, only the most frequently occurring parameter is considered and that is the time of the support phases.

Furthermore, this study shows that shorter double-support times while turning, lead to longer throwing distances. This is confirmed by recent investigations [6,25,26,27,28]. According to Hirose et al. [6], this would contradict statements of previous publications that the DS for each turn should be designed as long as possible [29,30].

For the velocities of the hands at the drop time points we determined, the results differ greatly across athletes. The calculated parameters from the left hand show significant correlations more often than the right hand. This is surprising, since no plausible explanation can be given. In turn, angular velocities at the time of throwing show almost no correlations with the achieved throwing distance and do not seem to be relevant for performance. In previous scientific work, no statements have been made about hand velocities. Further works could address this.

The PCA reveals that 20 of the 37 parameters load significantly on the first three main components. Each of them include a mixture of variables from durations (times), velocities and joint angles from different rotations in the turn phase. However, the principal components with their variables provide assignments that are statistically significant; but unfortunately, this makes an interpretation of the content impossible. Therefore, the results of the PCA are interpreted to the effect that variables, which load significantly on one of the principal components, are to be treated as parameters relevant and interesting for the technique of hammer throwing. The advantage of this approach is to obtain generally valid statements for the technique of the hammer throw. This type of analysis also gives an indication that stance times and the velocity of the left hand are relevant for a higher performance. If this can be confirmed by further studies an easy-to-use sensor-based system could be used in training practice. It would allow the coach and athlete to obtain quick feedback for improving the training.

## 5. Limitations

This study is limited, because only six German subjects were available. Furthermore, the performance of these athletes is quite far from world class. This could possibly be improved by using a wireless system [31]. It cannot be fully clarified whether these results are general correlations or individual ones. This study could only show tendencies and compare them with the already existing literature.

Another limitation of this experimental equipment is the explicit kinematic information of the hammer. This would allow a closer look at the influence of the athlete on the hammer and has the potential to reveal something new.

## 6. Conclusions

The study shows that a full-body sensor suit is able to capture all literature-based biomechanical parameters of the hammer throwing technique. Individual and general observations revealed significant correlations with stance times and left-hand velocity at different movement phases. However, we also demonstrated that each athlete has its specific profiles, leading to different results. Further studies are needed to extend the data tool with more athletes and more trials. In future, the results of such studies should be implemented in a slim version of a sensor-based system that can be integrated into everyday training.

## 7. Disclosure Statement

No potential conflict of interest was reported by the authors. This project (ZMVI4-070802/20-21) was funded with research funds from the Federal Institute for Sports Science based on a resolution of the German Bundestag. We would also like to express our thanks for the intensive talks and discussions with the national coach Mr. Helge Zöllkau, as well as the athletes who made themselves available for the investigations.

## Figures and Tables

**Figure 1 ijerph-19-05402-f001:**
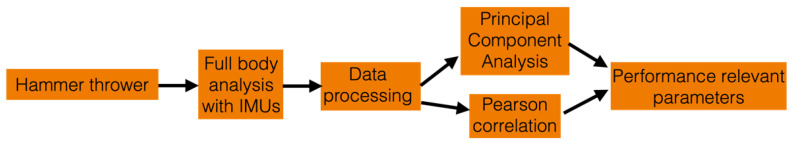
Flow-chart diagram for the main structure of the paper.

**Figure 2 ijerph-19-05402-f002:**
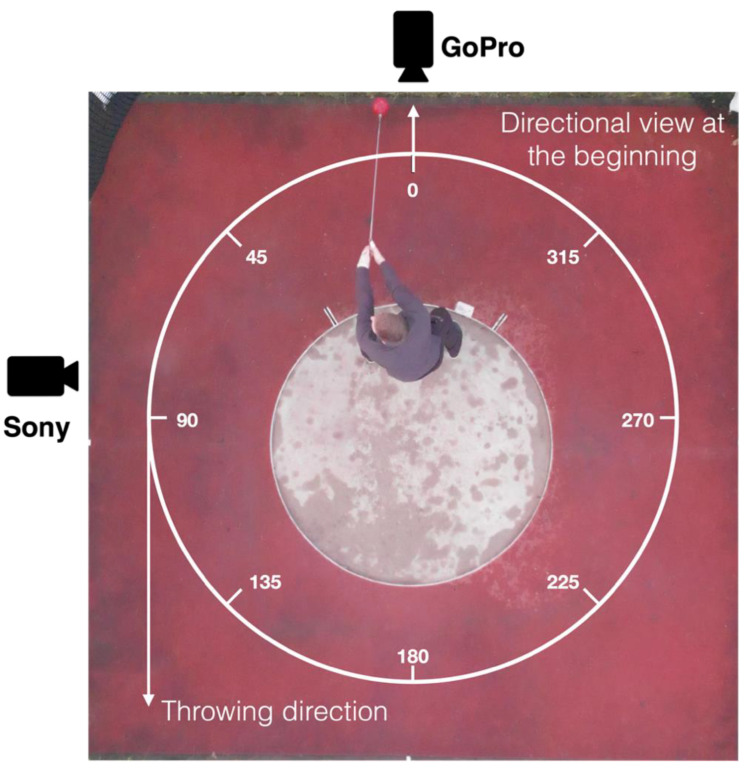
Study setup together with two video cameras. The division of the throwing circle in azimuth angles (°) is displayed.

**Figure 3 ijerph-19-05402-f003:**
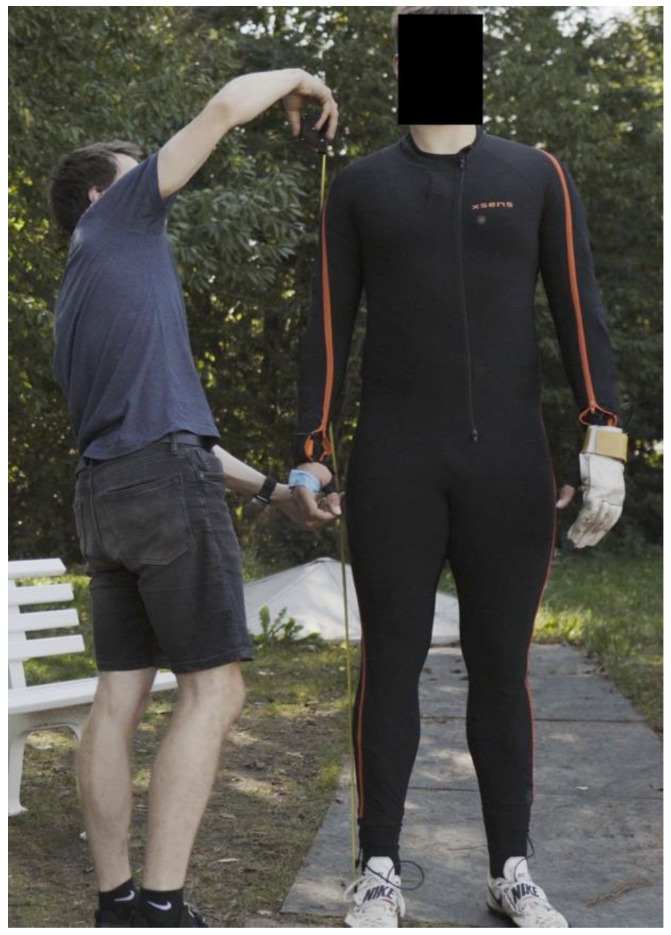
Anthropometric measurements of the athlete with Moven suit.

**Figure 4 ijerph-19-05402-f004:**
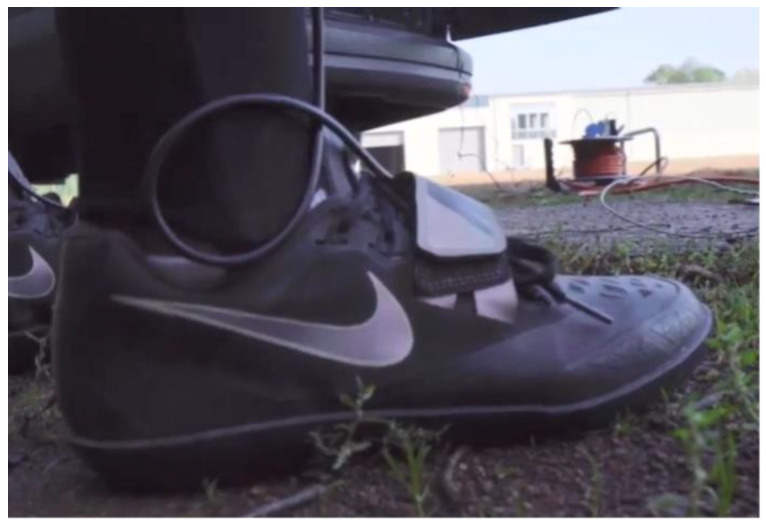
IMU placement under the Velcro fastener of the shoe.

**Figure 5 ijerph-19-05402-f005:**
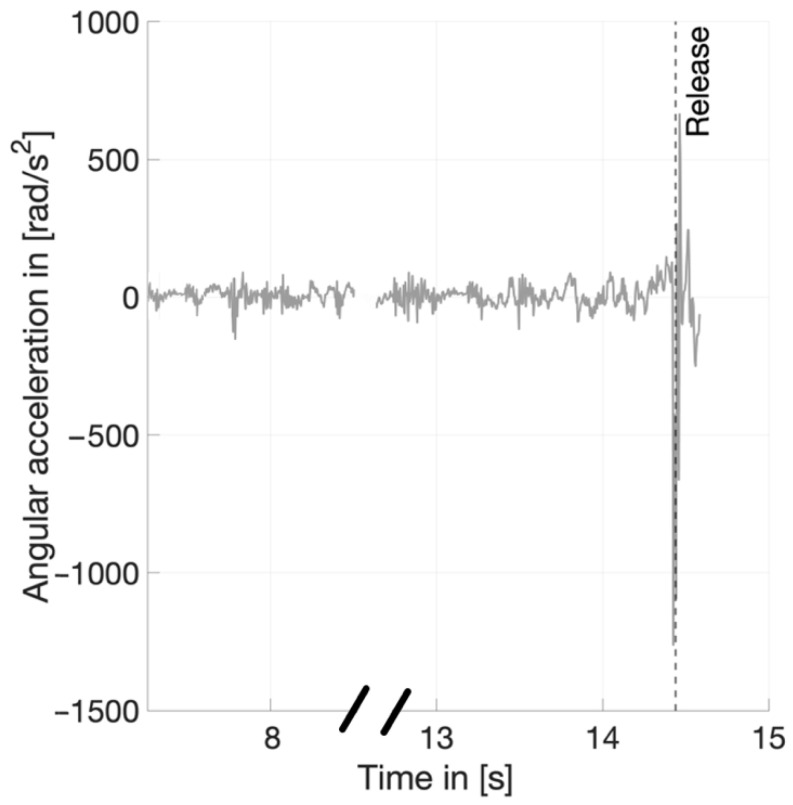
Determination of release frame (marked by dotted line) by means of the of angular acceleration course of the right hand.

**Table 1 ijerph-19-05402-t001:** Participating athletes’ characteristics, the number of trials evaluated (trials) and related mean throwing distance.

Athlete ID	Gender	Age (Years)	Mass of the Hammer (kg)	Body Mass (kg)	Body Height (m)	Trials	Mean Throwing Distance (m), STD
1	m	27	7.26	100	1.85	14	57.55 ± 3.05
2	f	30	4	72	1.67	17	56.34 ± 1.47
3	f	18	3	62	1.69	12	66.59 ± 1.73
4	f	20	4	57	1.67	12	49.37 ± 2.21
5	m	19	6	95	1.93	15	69.6 ± 8.73
6	f	17	3	60	1.70	12	55.1 ± 2.11

Note. M—male, f—female.

**Table 2 ijerph-19-05402-t002:** List and definitions of parameters used for determination of double- and single-support phases and for data.

Parameter	Definition/Calculation
**Temporal parameters for determining support phases**	
$\mathrm{Release}\;\mathrm{time}\;t_{release}$	Global maximum of the resulting angular Acceleration of the right hand
$\mathrm{Lift}\;\mathrm{of}\;\mathrm{the}\;\mathrm{circumferential}\;\mathrm{foot}\;\mathrm{within}\;\mathrm{the}\;\mathrm{turns}\;t_{lift,i};\;i=1\ldots j$	Positive and local intermediate maxima of the acceleration (z-axis) of the right toe
$\mathrm{Set}\;\mathrm{of}\;\mathrm{the}\;\mathrm{circumferential}\;\mathrm{foot}\;\mathrm{within}\;\mathrm{the}\;\mathrm{turns}\;t_{set,i};\;i=1\ldots j$	Positive and local maxima of the acceleration (z-axis) of the right toe
**Biomechanical parameters for technique analysis**	
$\mathrm{Duration}\;\mathrm{of}\;\mathrm{turns}\;t_{D}$ [s]	$t_{D}=t_{DS}+t_{SS}$
$\mathrm{Duration}\;\mathrm{of}\;\mathrm{single}-\mathrm{support}\;\mathrm{phases}\;t_{SS}$ [s]	$t_{SS}=\overset{j}{\underset{i=1}{\sum }}t_{SS,i}$
$\mathrm{Duration}\;\mathrm{of}\;\mathrm{double}-\mathrm{support}\;\mathrm{phases}\;t_{DS}$ [s]	$t_{DS}=\overset{j}{\underset{i=1}{\sum }}t_{DS,i}$
$\mathrm{Duration}\;\mathrm{of}\;\mathrm{individual}\;\mathrm{single}-\mathrm{support}\;\mathrm{phase}\;t_{SS,i}$ [s]	$t_{SS,i}=\frac{t_{set,i}-t_{lift,i}}{240\;\mathrm{Hz}},\;i=1\ldots j$
$\mathrm{Duration}\;\mathrm{of}\;\mathrm{individual}\;\mathrm{double}-\mathrm{support}\;\mathrm{phases}\;t_{DS,i},\;t_{DS,j}$ [s]	turn 1…j−1 $:\;t_{DS,i}=\frac{t_{lift,i}-t_{set,i}}{240\;\mathrm{Hz}}$ , i=1…j−1 turn j $:\;t_{DS,j}=\frac{t_{release}-t_{set,j}}{240\;\mathrm{Hz}}$
Velocity of the left hand at release [m/s]	$\mathrm{Resulting}\;\mathrm{velocity}\;\mathrm{of}\;\mathrm{the}\;\mathrm{left}\;\mathrm{hand}\;\mathrm{at}\;\mathrm{frame}\;t_{release}$
Velocity of the right hand at release [m/s]	$\mathrm{Resulting}\;\mathrm{velocity}\;\mathrm{of}\;\mathrm{the}\;\mathrm{right}\;\mathrm{hand}\;\mathrm{at}\;\mathrm{frame}\;t_{release}$
Angular velocity of the left hand at release [rad/s]	$\mathrm{Resulting}\;\mathrm{angular}\;\mathrm{velocity}\;\mathrm{of}\;\mathrm{the}\;\mathrm{left}\;\mathrm{hand}\;\mathrm{at}\;\mathrm{frame}\;t_{release}$
Angular velocity of the right hand at release [rad/s]	$\mathrm{Resulting}\;\mathrm{angular}\;\mathrm{velocity}\;\mathrm{of}\;\mathrm{the}\;\mathrm{right}\;\mathrm{hand}\;\mathrm{at}\;\mathrm{frame}\;t_{release}$

Note. Descriptions refer to a thrower using *j* turns, holding the hammer with his/her left hand, turning counterclockwise and using the right leg as the circumferential leg during single-support phase. Frame describes the individual measuring points of the IMU.

**Table 3 ijerph-19-05402-t003:** Pearson correlation coefficients for main parameters.

Parameter	Athlete ID					
	1	2	3	4	5	6
duration of turns	−0.78 *	−0.84 *	−0.38	−0.77 *	−0.64 *	−0.30
duration DS	−0.55 *	−0.55 *	−0.53	−0.88 *	−0.65 *	−0.49
duration SS	−0.55 *	−0.57 *	0.24	0.10	0.02	0.37
duration DS1	−0.46	−0.62 *	−0.11	−0.311	−0.40	−0.53
duration DS2	−0.52	0.04	−0.55	−0.50	−0.82 *	−0.20
duration DS3	−0.20	−0.30	−0.25	−0.44	−0.33	−0.14
duration DS4	−0.38	−0.38	0.01	−0.47	−0.48	-
duration SS1	−0.44	−0.31	0.21	0.42	−0.61	0.42
duration SS2	−0.26	−0.67 *	−0.06	0.79 *	−0.46	0.26
duration SS3	−0.45	−0.51	−0.16	−0.15	0.63	0.21
duration SS4	−0.50	−0.05	0.31	−0.65	0.54	-
release velocity left hand	0.52	0.70 *	0.25	0.68 *	0.65 *	−0.11
release velocity right hand	0.14	0.23	0.36	0.58	0.63 *	0.51
release angular velocity left hand	0.60 *	0.16	0.10	−0.15	0.58	0.45
release angular velocity right hand	−0.02	−0.07	0.24	−0.10	−0.67 *	0.29

Note. *, Correlation is significant.

**Table 4 ijerph-19-05402-t004:** Results of principal component analysis.

Principal Component 1	Principal Component 2	Principal Component 3
duration SS1	xfactor at lift 2–4	xfactor at set 3
durations DS1 + 2	duration DS4	xfactor at lift 1
flexion angles of the left knee at set 1 and 4	duration SS4	durations SS2 + 3
maximum velocity of the left hand in DS1	maximum velocities of the left hand in DS2 + 3	loss of velocity of the left hand between DS2 and SS3
differences in pelvis height between DS3 and SS4 and between SS4 and release	velocities of the left and right hand at release	
	minimal flexion angle of right hip in SS4	

Note. KMO test (0.73) and Bartlett test (*p* < 0.001) are both fulfilled. Kaiser criterion and Scree plot justified the extraction of three principal components with eigenvalues ≥ 1 and describes 66.48% of variance of the system. The system was rotated using Varimax method.

## Data Availability

Data supporting reported results can be requested by the author Stefan Tiedemann (stefan.tiedemann@ovgu.de).

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
