# Peer review of "Application of Inertial Sensors to Identify Performance-Relevant Parameters in Olympic Hammer Throw"

_ijerph, 2022, doi:10.3390/ijerph19095402_

Round 1
Reviewer 1 Report
The paper contains a valuable contribution. The subject is within the scope of the journal and the objective of research is well stated. However, some clarifications about the underlying hypothesis / scope are needed.
In the opinion of this Reviewer the manuscript deserves to be published once the Author takes into account the raised issues.
Introduction / Literature review
- The research scope is clear as well as the literature review. Anyway, the authors should better highlight the innovative aspects of their work in the manuscript.
What are the advantages / findings in the proposed paper, which are not covered by other studies/reviews?
- Please add some reference to the MVN. The authors call it Moven Suit, but on Xsens technologies website “MVN Animate” and “MVN Analyze” are present.
- For the sake of readability, at the end of Section 1 the authors should describe how the paper is structured.
Materials and methods
- A very important part that is missing in the paper is how the system works. No details on how the data are sampled and collected (filtering, full scale range, data concentrator, data fusion, accuracy, etc.) are given. Only the sampling rate is reported. Where the samples are collected? Are the values collected in real-time or is it an off-line system?
- Row 132: the authors use the angular acceleration of the right hand. How is this parameter calculated? Which data are extracted from the MVN suit and which parameter is calculated in matlab? In both cases, how are the parameters calculated? Please give more detail about it.
- Row 157: Only 57 throws with 4 turns were used. So it seems that not only the athlete #6 (as reported in table 3) with 12 trials (as reported in table 1) did not use four turns, but that other athletes used four or less turns. Is that correct? In any case, please clarify this aspect.
- Table 2: it is a subset of table 5. The authors can use only table 5.
Results
- Row 204: table 4 is not in the appendix
- Table 4: it is a subset of table 6. The authors can use only table 6.
- Table 4 and table 6: please specify the unity of measure
Discussion and conclusions
- The section conclusions is repeated in section 4 and 6.
- Row 222: the authors say that previous works used probably data not accurate enough. Which is the accuracy of the system used in this paper?
- Row 228: the authors talk about the importance of event detection timing of the system in relation to the sampling frequency. Actually in such systems where different boards collect different data, the key in the synchronization. Synchronization is an important issue in such a system and a lot of papers analyze it in detail (e.g. , https://doi.org/10.23919/SASE-CASE.2017.8115371, https://doi.org/10.1109/ICSPCT.2014.6884917).
How much does synchronization problem could affect the entire system? I assume that in order to correlate different measures coming from different acquisition system, the timestamp of the acquisition is crucial.
Minor
- The authors should check that all the used acronyms are not repeated (e.g. IMU in the abstract, PCA, etc)
- Mainly the English is good and there are only a few typos. However, the paper should be carefully rechecked.
- Please specify the unity of measurement
Author Response
- The research scope is clear as well as the literature review. Anyway, the authors should better highlight the innovative aspects of their work in the manuscript. (We corrected it in the abstract (Line: 7-19))
- What are the advantages / findings in the proposed paper, which are not covered by other studies/reviews? (We have a closer look to kinematics like acceleration and angular velocity of several body segments (Line: 109))
- Please add some reference to the MVN. The authors call it Moven Suit, but on Xsens technologies website “MVN Animate” and “MVN Analyze” are present. (Done (Line: 79 and84))
- For the sake of readability, at the end of Section 1 the authors should describe how the paper is structured. (We added a flow-chart-diagram in the introduction as a general overview (Line: 88))
- A very important part that is missing in the paper is how the system works. No details on how the data are sampled and collected (data concentrator, data fusion, accuracy, etc.) are given. Only the sampling rate is reported. Where the samples are collected? Are the values collected in real-time or is it an off-line system? (We add this part (Line: 79-84 and 108-112))
- The authors use the angular acceleration of the right hand. How is this parameter calculated? Which data are extracted from the MVN suit and which parameter is calculated in matlab? In both cases, how are the parameters calculated? Please give more detail about it. (The data is directly from the sensor (Line: 145))
- Only 57 throws with 4 turns were used. So it seems that not only the athlete #6 (as reported in table 3) with 12 trials (as reported in table 1) did not use four turns, but that other athletes used four or less turns. Is that correct? In any case, please clarify this aspect.(This is correct and we made it more clear (Line: 166-170))
- Table 2: it is a subset of table 5. The authors can use only table 5. (We changed it (Line: 189))
- Line 140: Please correct Table number (Done (Line: 153))
- Line 204: table 4 is not in the appendix (We corrected this mistake (Line: 213))
- Table 4: it is a subset of table 6. The authors can use only table 6. (Table 4 is the result of the PCA and not in the appendix)
- Table 4 and table 6: please specify the unity of measure (These tables have no units because these are the results from the Pearson correlation)
- Row 222: the authors say that previous works used probably data not accurate enough. Which is the accuracy of the system used in this paper? (We added the specific of time accuracy (Line: 243))
- Row 228: the authors talk about the importance of event detection timing of the system in relation to the sampling frequency. Actually in such systems where different boards collect different data, the key in the synchronization. Synchronization is an important issue in such a system and a lot of papers analyze it in detail (e.g. , https://doi.org/10.23919/SASE-CASE.2017.8115371, https://doi.org/10.1109/ICSPCT.2014.6884917). (We specify the aspect of synchronization (Line: 244))
- How much does synchronization problem could affect the entire system? I assume that in order to correlate different measures coming from different acquisition system, the timestamp of the acquisition is crucial. (With the wired version of the MVN Link system synchronisation is no problem (Line: 243-245). With the Awinda system, which works wirelessly, this is a potential problem.)
- The authors should check that all the used acronyms are not repeated (e.g. IMU in the abstract, PCA, etc) (We corrected it)
- Mainly the English is good and there are only a few typos. However, the paper should be carefully rechecked. (We rechecked it)
- Please specify the unity of measurement (We added it)
Reviewer 2 Report
Thank you very much for sending the article titled: Application of inertial sensors to identify performance-relevant parameters in Olympic hammer throw. Generally, the paper is quite interesting to my mind, however, the authors should refer to the following statements:
- please rewrite Abstract (Background, Objectives, Methods, Conclusion)
- please include a Flowchart in the beginning section Materials and Methods
- to improve the quality of Figure 4, I suggest presenting results in the broken axis. Moreover, the author present this figure and wrote: Figure 4 shows a typical time course angular acceleration of the right hand for an exemplary throw. It is on experiment result?
- The authors used a Moven Suit with Xens software. Please write parameters Xens system.
- line 140, please correct Table number (there should be Table 6A)
- The Moven suit can restrict the athlete's movement through the transitions between the sensors. I suggest using wirelless IMUs. For example in the article titled: Dynamic Model of a Humanoid Exoskeleton of a Lower Limb with Hydraulic Actuators, Sensors 2021, authors used IMU to obtain kinematic parameters. This kind of IMU is more flexible. To improve article quality, please refer to this publication in a few sentences (in the Limitation of the study section)
- why duration of turn was statistically imporant for 1,2,4,5, wheras for 3 and 6 not?
Author Response
- Please rewrite Abstract (Background, Objectives, Methods, Conclusion) (Line: 7-20)
- Please include a Flowchart in the beginning section Materials and Methods (Line: 89)
- To improve the quality of Figure 4, I suggest presenting results in the broken axis. Moreover, the author present this figure and wrote: Figure 4 shows a typical time course angular acceleration of the right hand for an exemplary throw. It is on experiment result (it is a pre-test)? (Done (Line: 149))
- The authors used a Moven Suit with Xens software. Please write parameters Xens system. (Line: 109-112)
- Line 140, please correct Table number (there should be Table 6A) (done)
- The Moven suit can restrict the athlete’s movement through the transitions between the sensors. I suggest using wireless IMUs. For example in the article titled: Dynamic Model of a Humanoid Exoskeleton of a Lower Limb with Hydraulic Actuators. Sensors 2021. authors used IMU to obtain kinematic parameters. This kind of IMU is more flexible. To improve article quality, please refer to this publication in a few sentences in the Limitation of the study section. (The athletes says that the movement was not restricted by the sensors. All the wires are clever implemented in the suit. We think that this aspect should be added (Line: 294) but we believe that the negative sides of the wireless system do not outweigh the advantages (problem with synchronisation, lower sample rate (60 vs. 240 Hz) and lower range).
- Why duration of turn was statistically imporant for 1,2.4,5, wheras for 3 and 6 not? (We think it has to do with the individual aspects of athletes)